# Design and flight results of the VHF/UHF communication system of Longjiang lunar microsatellites

Mingchuan Wei [1], Chaoran Hu [1], Daniel Estévez [2], Mier Tai[1], Yuhao Zhao [1], Jiahe Huang [1], Cees Bassa [3,4], Tammo Jan Dijkema [3,4], Xibin Cao [1✉] & Feng Wang [1✉]

As a part of China's Chang'e-4 lunar far side mission, two lunar microsatellites for low frequency radio astronomy, amateur radio and education, Longjiang-1 and Longjiang-2, were launched as secondary payloads on 20 May 2018 together with the Queqiao L2 relay satellite. On 25 May 2018, Longjiang-2 successfully inserted itself into a lunar elliptical orbit of 357 km × 13,704 km, and became the smallest spacecraft which entered lunar orbit with its own propulsion system. The satellite carried the first amateur radio communication system operating in lunar orbit, which is a VHF/UHF software defined radio (SDR) designed for operation with small ground stations. This article describes and evaluates the design of the VHF/UHF radio and the waveforms used. Flight results of the VHF/UHF radio are also presented, including operation of the radio, performance analysis of downlink signals and the first lunar orbit UHF very-long-baseline interferometry (VLBI) experiment.

[1] Harbin Institute of Technology, No. 92 Xidazhi Street, 150001 Harbin, China. [2] Oceano Atlantico 38, Tres Cantos, 28760 Madrid, Spain. [3] C.A. Muller Radio Astronomy Station, Oude Hoogeveensedijk 4, 7991 PD Dwingeloo, The Netherlands. [4] ASTRON, Netherlands Insititute for Radio Astronomy, Oude Hoogeveensedijk 4, 7991 PD Dwingeloo, The Netherlands. ✉email: xbcao@hit.edu.cn; wfhitsat@hit.edu.cn

Longjiang-1 and Longjiang-2 are a pair of lunar micro-satellites for low-frequency radio astronomy, amateur radio, and education, developed by Harbin Institute of Technology, as a part of the Chang'e-4 lunar far side mission[1]. The satellites, each with a volume of 765 × 420 × 570 mm³ and a mass of ~47 kg, were launched into a lunar transfer orbit on 20 May 2018 by a CZ-4C rocket as secondary payloads, together with the Queqiao L2 relay satellite. Unfortunately, Longjiang-1 was lost because of a malfunction of thruster control logic during the first trajectory correction maneuver (TCM). The logic on Longjiang-2 was later patched, and after 113 h of flight since launch, the satellite successfully inserted itself into a lunar elliptical orbit of 357 km × 13704 km, and became the smallest spacecraft which entered lunar orbit independently.

Besides the S-band and X-band radios operated by the Chinese Deep Space Network (CDSN), Longjiang-1 and Longjiang-2 were equipped with identical VHF/UHF radios for amateur radio experiments and backup tracking, telemetry, and command (TT&C). The radio onboard Longjiang-2 is the first radio communication system operating on amateur radio bands that was placed into lunar orbit.

Radio amateurs have been building and tracking satellites since 1961[2]. Nowadays, many small satellites developed for universities and technology development projects use amateur radio frequencies and AX.25 link layer protocol[3,4]. With the development of the Internet, several global satellite tracking networks have been developed by radio amateurs, for example SatNOGS[5]. Besides that, radio amateurs around the word use the Moon as a natural reflector for moonbounce, or Earth–Moon–Earth (EME) communication. For these purposes, radio amateurs operate a large number of high gain antennas around the world, forming a possible non-governmental, non-commercial deep space network, though the size of antennas is relatively small. VHF/UHF is among the most popular bands used for EME.

Before Longjiang-2, several satellites operating on amateur radio bands have been launched into deep space, including UNITEC-1 (trans-Venus, Japan, 2010)[6], Shin'en2 (heliocentric orbit, Japan, 2014)[7], ARTSAT2-DESPATCH (heliocentric orbit, Japan, 2014)[8], and 4M-LXS (lunar flyby, Luxembourg, 2014)[9]. These satellites used CW or JT65 modulation schemes for data downlink because of the low demodulation thresholds of these modes, but the data rates are too low (<10 bps) for payload data, so only transmission of some very basic housekeeping data is possible. UHF frequencies, some of which are allocated to the amateur satellite service, are widely used by deep space missions for inter-probe proximity links, for example the link between China's Chang'e-3 lander and Yutu rover, and NASA's Electra proximity payload for Mars explorers[10].

In this paper, we report on the VHF/UHF communication system of the Longjiang lunar micro-satellites. As piggyback lunar micro-satellites, Longjiang-1 and Longjaing-2 were very restricted in many aspects. Each of the satellites has a wet weight of only 47 kg, including 15 kg of propellant for the necessary orbit maneuvers, so the mechanics and electronics must be very lightweight. Deployable solar panels were not used because of weight issues, which limits the capabilities for power generation. Antenna size is limited by the available envelope of the launch vehicle, making it difficult to achieve high gain and efficiency. Based on the analysis above, it was a challenge for the VHF/UHF radio to operate in lunar orbit with limited onboard and ground station resources. To overcome this, a low-power software defined radio (SDR) transceiver and new uplink and downlink waveforms were designed and successfully demonstrated in trans-lunar and lunar orbit.

## Results

**Mission analysis and system configuration.** Longjiang-1 and Longjiang-2 were designed to separate from the launch vehicle at an altitude of ~200 km over the Pacific. Before orbiting the Moon, the satellites had to perform several trajectory correction maneuvers (TCMs) and a maneuver for lunar orbit injection (LOI), with their own propulsion. After that, a few maneuvers were needed to achieve a stable elliptical orbit. Their partner, Queqiao relay satellite passed by the Moon and continued its journey to the L2 halo orbit[1]. Figure 1a shows a picture of the Longjiang-1/2 satellites on the final stage of CZ-4C launch vehicle, together with the Queqiao relay satellite[11]. The orbit of Longjiang-1/2 from Earth to the Moon and the positions of maneuvers are shown in Fig. 1b.

Based on the designed orbit, after orbiting the Moon, the distance between the satellite and a typical ground station ranges from ~340,000 km to ~420,000 km, much larger than for LEO satellites. The line-of-sight velocity between the satellite and a typical ground station ranges between about ±2 km/s, much smaller than for LEO satellites. Downlink and uplink budgets for the VHF/UHF radio is shown in Supplementary Table 1 and Supplementary Table 2.

The missions of the VHF/UHF radio are defined as:

(a) Lunar orbit amateur radio experiment. Downlink signals could be received with reasonable-sized antennas and commercial off-the-shelf receivers when the satellites were in lunar transfer orbit and lunar orbit.

(b) Backup telemetry and command, especially when S-band ground stations were not available. Particularly, after the satellites were deployed from launch vehicle, before S-band was available, first telemetries from the satellites were expected to be received on UHF.

(c) Provide power and control/data interface to a miniature CMOS color camera. Image data from the CMOS camera could be downloaded via UHF downlink.

(d) Provide an open command interface to allow radio amateurs to send commands to control the camera.

A conventional S-band or X-band satellite TT&C system typically uses two circular polarization antennas in opposite directions fed by an RF network consisting of circulators and hybrid couplers for an omni-directional coverage. But for VHF/UHF band, the size of circular polarization antennas and such an RF network is too large for Longjiang-1/2. On the other hand, VHF/UHF dual-band antennas are widely used by mobile communication systems. A simple duplexer consisting of a low-pass filter and a high-pass filter can be used to split the receiving and transmitting signals. A problem of VHF/UHF dual-band antennas is that they usually have linear polarization and deep nulls. To overcome this, two linear-polarization, dual-band shortened antennas were used and mounted in −X and +Z directions, as shown in Fig. 1c. These two antennas fill the nulls of each other, and result in an omni-directional coverage. The two antennas are connected to a pair of transceivers. The two receiving channels operate on the same frequency, and the two transmitting channels operate on two frequencies spaced by 1 MHz. For uplink, a command can be received by one or both of the transceivers. For downlink, the two transmitters usually work in burst mode to save power, and can be switched on simultaneously to increase available data rate, and get a wider bandwidth for VLBI measurement.

**Transceiver design.** The VHF/UHF radio design of Longjiang-1/2 includes two independent SDR transceivers and a miniature CMOS camera, integrated into a layer of the stack of onboard

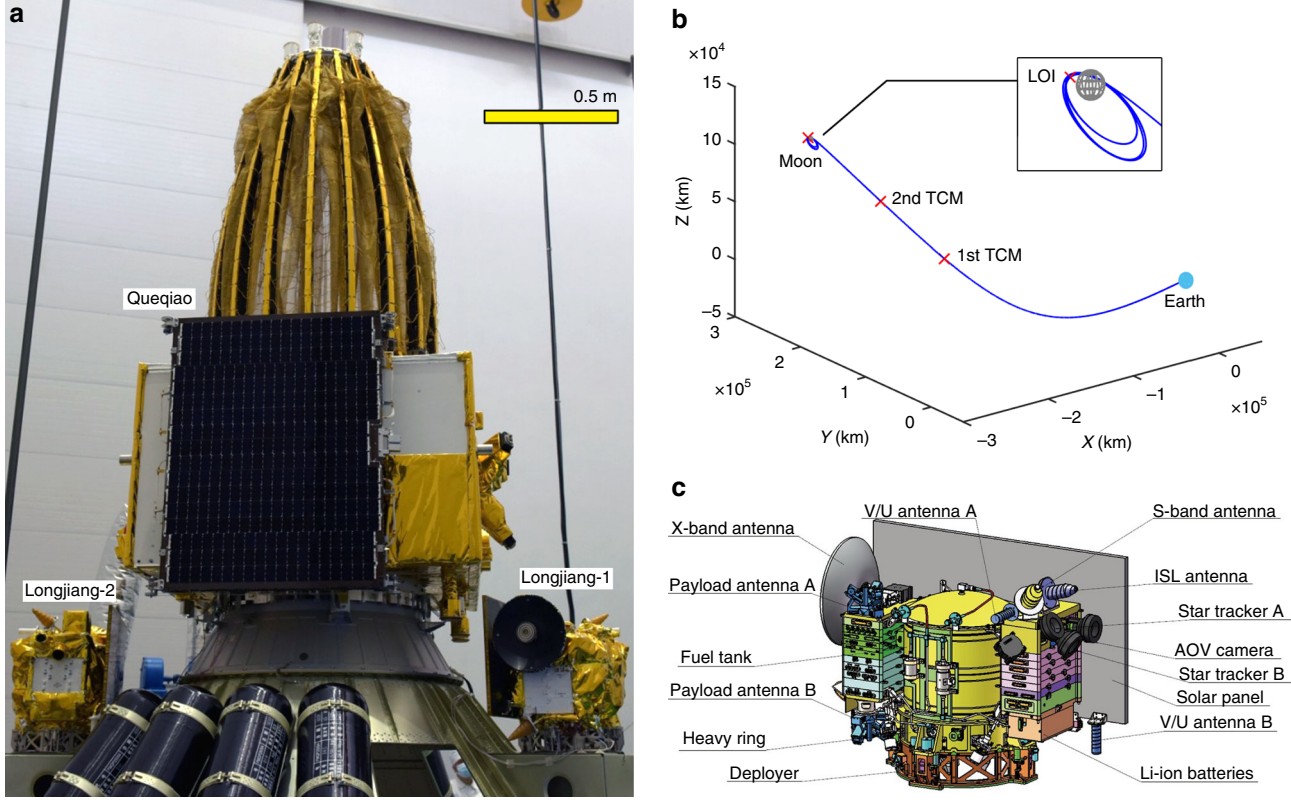

**Fig. 1 Longjiang-1/2 satellites. a** The Longjiang-1/2 microsatellites and Queqiao relay satellite on the final stage of the launch vehicle. **b** Orbit of Longjiang-1/2 from Earth to the Moon in moon-fixed reference frame. **c** Equipment arrangement of Longjiang-1/2.

electronics. Each transceiver includes a low intermediate frequency (LIF) I/Q receiver and a direct modulation transmitter. Digital baseband processing is done by an ARM processor. The transceivers can be reconfigured for different uplink and downlink waveforms, without modifications on hardware. The block diagram of the VHF/UHF radio is shown in Fig. 2a, pictures showing the flight hardware of VHF/UHF radio and antenna are shown in Fig. 2b, c.

Many SDRs use an FPGA for signal processing, making it a complex and large power consuming system. For low data rate applications, using processors for signal processing is also possible. The satellite ARISSat-1, developed by AMSAT, had an onboard SDR transponder based on dsPIC processors and was launched in 2011[12]. Harbin Institute of Technology also developed a series of SDR transceivers based on ARM Cortex-M4F processors for LilacSat-2, BY70-1, LilacSat-1[13], etc. The unit onboard LilacSat-2 has been operating in LEO for more than 4 years. For Longjiang-1/2, an ARM Cortex R4F processor with lockstep CPUs and EDAC protected memories was selected for the radiation environment in lunar orbit.

In the receiver path, the 145-MHz input signal is first amplified by a low noise amplifier, then down-converted to 98 kHz intermediate frequency (IF) by an image rejecting I/Q demodulator, then filtered and amplified by the IF filter and amplifier which also convert the differential signal to single ended. Finally, the I and Q signals are digitized by a dual channel A-D converter at a sample rate of 56 ksps. No analog auto gain control (AGC) was used for the receiver path, and the gains of amplifiers were set as low as possible. This improved the performance for a very weak burst uplink. Floating point operation is used in demodulators to provide sufficient dynamic range.

The transmitting path is quite simple. An FSK/GMSK modulator directly modulates the data to the 435-MHz

transmitter carrier. Then the modulated signal is amplified by a class A driver amplifier, then by a high-efficiency class C power amplifier. A temperature sensor is used to monitor the temperature of the power amplifier. The output signal is filtered to suppress emissions on receiver and harmonic frequencies, before the port is combined with the receiver using a duplexer.

The transceivers also provide power and data interface to the miniature CMOS camera. Both the transceivers can take control of the camera.

Specifications of the VHF/UHF radio are shown in Supplementary Table 3.

**Waveform design**. Two downlink waveforms and two uplink waveforms are designed for the Longjiang-1/2's VHF/UHF radio: GMSK telemetry, JT4G telemetry, GMSK telecommand, and low rate telecommand (LRTC).

The telemetry link is the most critical link of Longjiang-1/2's VHF/UHF radio. GMSK modulation was selected because of several advantages:

(a) Continuous phase. GMSK is a continuous phase modulation. The output stage of the transmitter can use a class C amplifier.
(b) Good bit error rate (BER) performance. When using a coherent demodulator, a BER performance quite similar to BPSK/QPSK can be achieved, especially when BT = 0.5.
(c) Simple modulator hardware. GMSK signals can be generated from several methods, including with an OQPSK modulator or a simple FM modulator, even by direct control of the frequency control word of a VCO.
(d) High spectrum efficiency. GMSK is one of the modulations that was recommended by CCSDS for medium rate telemetry, mainly because of its high spectrum efficiency.

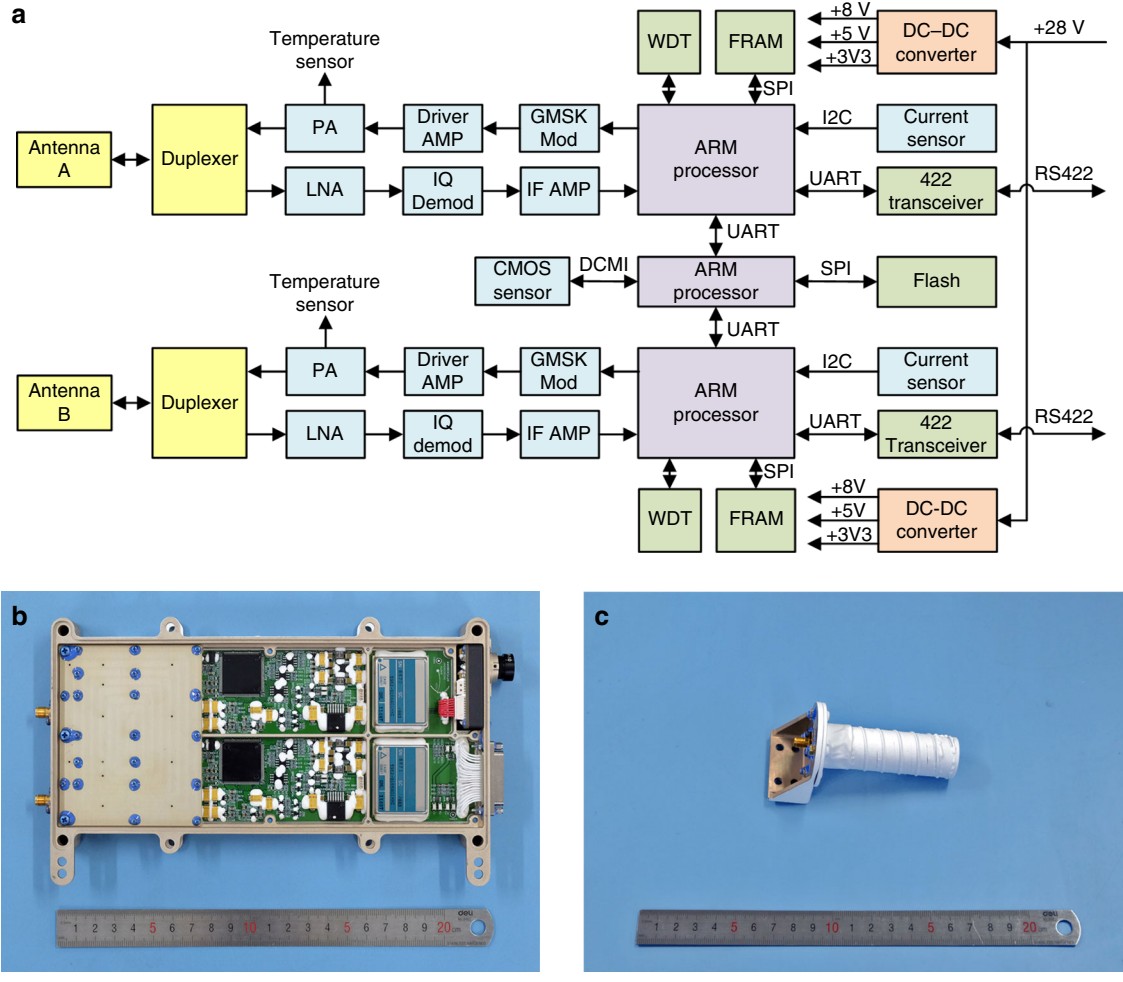

**Fig. 2 Hardware of VHF/UHF radio. a** Block diagram of VHF/UHF radio. **b** Flight hardware of the VHF/UHF transceivers and miniature CMOS camera. **c** Flight hardware of a VHF/UHF dual-band antenna.

However for Longjiang-1/2, bandwidth is not a main concern because the data rate is quite low.

GMSK telemetry can operate in burst mode to transmit basic status of the satellite and the radio itself as a bacon, or stream mode to transmit detailed housekeeping of all the subsystems. The symbol rate can be switched between 250 and 500 bps. Turbo code is selected for channel coding, as it provides the highest coding gain among the codes recommended by CCSDS. The block size selected is 1784, and coding rate can be switched among 1/2, 1/3, 1/4, and 1/6.

On the ground side, variations on offset quadrature phase shift keying (OQPSK) receivers can be used for GMSK demodulation. S. Shambayati and D. K. Lee provided the FER results of the standard DSN OQPSK receiver and some of its variations for medium rate telemetry[14]. For the case of Longjiang-1/2, the situation is more difficult. The data rate and signal $C/N_0$ for Longiang-1/2 is quite low, so a narrow loop filter has to be used. The resulting acquisition time of the carrier tracking loop is too long for burst mode operation, which is used to reduce power consumption. To improve the system performance, an attached synchronization marker (ASM) detector, which acts as a correlator in both time and frequency domain, is used to aid the acquisition of the carrier tracking loop, as shown in Fig. 3a. The input stream is first multiplied by a set of taps, which are the conjugates of the ASM, before calculating an FFT, then the FFT output bin with maximum power is searched in both time and

frequency domain. The power of the bin is used for an open loop auto gain control (AGC) and estimation of $E_b/N_0$, which is needed by the turbo decoder. The frequency and phase of the bin is used to set the initial state of NCO. The time when the maximum power is found is marked for symbol timing.

An OQPSK carrier tracking loop is used for carrier recovery. A matched filter, which is the first order Laurent decomposition of $BT = 0.5$ GMSK signal, is also included for bit sharpening[15]. The output of the matched filter is then sampled at the proper time to generate the demodulated symbols.

To work with the coherent demodulator, a precoder is used to avoid the propagation of bit errors introduced by the inherently differential property of GMSK modulation as recommended by CCSDS[16].

The demodulators were implemented with C++ and Python on GNU Radio. The resulting $C/N_0$ threshold for 500 baud GMSK with $r = 1/4$ turbo code or 250 baud GMSK with $r = 1/2$ turbo code with a packet error rate of 0.1 is ~24 dBHz. With a good low noise amplifier (LNA), a sensitivity of ~−149 dBm can be achieved.

The JT4G beacon was designed to be capable of being received by very small ground stations and included some very basic status of the radio for trouble shooting purposes. The mode is selected from the modes designed by Joe Taylor for EME communications[17]. The most popular mode among these is JT65B, which uses 2.69 bps 65-FSK with (63, 12) Reed-Solomon code, and was used by 4M-LXS lunar flyby mission. For Longjiang-1/2, as it was

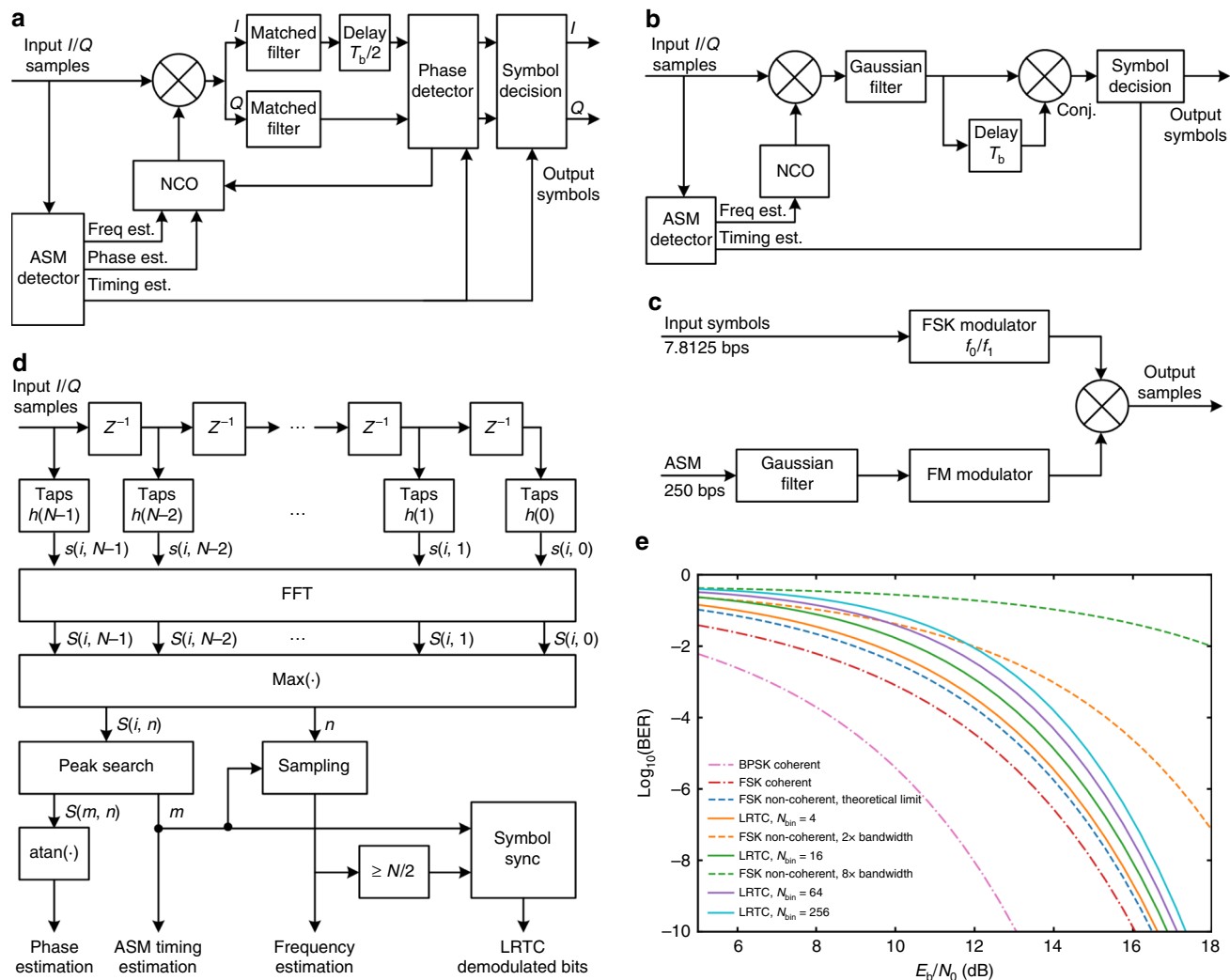

**Fig. 3 Modulators and demodulators of the VHF/UHF communication system. a** Block diagram of ground station GMSK coherent receiver. **b** Block diagram of onboard GMSK telecommand receiver. **c** Block diagram of ground station LRTC transmitter. **d** Block diagram of onboard LRTC receiver. **e** BER performance of LRTC demodulator compared with some other demodulators.

designed to orbit the Moon in an elliptical orbit, the Doppler rate at perigee would be quite large. For this reason, the JT4G mode was selected, which uses 4.375 bps 4FSK for better robustness for larger Doppler spread and Doppler rate. An $r = 1/2$, $k = 32$ convolutional code is used by JT4G for channel coding. The resulting $C/N_0$ threshold is ~17 dBHz.

GMSK is also the modulation selected for telecommand uplink. The symbol rate is 250 bps. For robustness and onboard simplicity, a non-coherent demodulator was developed based on a complex version of the one-bit differential detector introduced by M. K. Simon[18]. The input signal is first filtered by a Gaussian filter, then divided into two arms. One arm is delayed by one symbol time then conjugated, and another not changed. Then the two arms are multiplied together, then sampled to get the recovered symbols. The ASM detector described before is also used to provide frequency and timing estimation. A (64, 32) Reed-Solomon code is used for channel coding. The resulting $C/N_0$ at threshold is ~33 dBHz and the receiver sensitivity is −132 dBm. A block diagram of the onboard GMSK telecommand receiver is shown in Fig. 3b.

A new LRTC system was developed for better sensitivity than GMSK telecommand. The idea is to represent the symbols by transmitting GMSK-modulated ASMs on two different

frequencies. It can be regarded as FSK-DSSS-GMSK, though the frequency shift of GMSK is narrower than the frequency shift of FSK. Alternatively, it can be considered that the modulated ASM is used as the symbol shaping of FSK symbols. This results in a symbol rate of 7.8125 bps. Figure 3c shows the block diagram of the low rate telecommand transmitter.

Onboard the satellite, the ASM detector for GMSK telemetry is reused as a frequency discriminator. The ASM detector is a correlator for a pseudo-random sequence. It is known that this introduces processing gain that attenuates several kinds of interfering signals. Figure 3d shows the block diagram of onboard low rate telecommand receiver. Figure 3e shows the BER performance of LRTC demodulator with different values of $N_{bin}$, compared with some other modulations and demodulators, where $N_{bin}$ is the number of FFT bins used to search for maximum power. When $N_{bin} = 2$, the bandwidth and BER performance of LRTC demodulator is identical to the theoretical limit of non-coherent FSK demodulator. Indeed, the LRTC demodulator is a special kind of non-coherent FSK demodulator. When the symbol rate is very low, a much larger frequency shift and receiver bandwidth than the symbol rate is usually used to achieve better tolerance of frequency error. In this case, the LRTC demodulator provides much better BER performance than the

typical non-coherent FSK demodulator, and is more robust against narrowband interference.

**Operation of the VHF/UHF radio**. The VHF/UHF radios onboard Longjiang-1 and Longjiang-2 were powered on as soon as the satellites were separated from the launch vehicle at 20 May 2018 21:54:50 UTC and 20 May 2018 21:55:20 UTC. Radio amateurs in Brazil, Chile, and the US spotted the downlink signals of both satellites, and kept tracking the satellites until 21 May 2018 02:49 UTC, when the VHF/UHF radios were powered off because of overheating of the batteries. Meanwhile the maximum distance from the satellites to the ground stations was <70,000 km, so just a small antenna was required for receiving. The radios were operating in burst mode, and transmitted the housekeeping parameters of the satellite buses and the radios themselves every 5 min. 37 packets were received from Longjiang-1, and another 37 packets from Longjiang-2.

On 23 May 2019, the VHF/UHF radio of Longjiang-2 was switched on after the first course correction, from UTC 12:31 to 12:55. Signals from the satellite was received by the 12-m dish antenna in Shahe, China and radio amateurs in Poland.

Longjiang-2 achieved a lunar elliptical orbit of $357 \times 13704$ km after a successful lunar orbit insertion at 25 May 2018 14:08 UTC. The first activation of the VHF/UHF radio after orbiting the Moon was from 2 June 2018 22:00:00 UTC to 2 June 2018 23:50:00 UTC. Downlink signals from the satellite were received in the Netherlands, Poland, UK, and China. This was the first transmission on amateur radio bands from lunar orbit.

The first image transmission via the VHF/UHF radio was on 4 August 2018, to download an image of starry sky with Mars in view. The VHF/UHF radio provided the team with a direct link to control the onboard miniature CMOS camera, allowing the satellite to respond to some occasional imaging tasks. Figure 4c shows an image of total solar eclipse in South America taken by Longjiang-2, with the Moon, Earth, and eclipse shadow in view. The image was taken by the miniature CMOS camera on 2 July 2019, and transmitted via the VHF/UHF radio using slow scan digital video (SSDV) format[19] on 3 July 2019. The file size of the image is 19.1 kbytes and took ~22 min to download at 500 baud with $r = 1/4$ turbo code.

On 7 October 2018, the UHF downlink signals bounced off the Moon were first observed. The direct path and the moonbounce path signals are distinguished by different Doppler, and match the prediction quite well, as shown in Fig. 4f. The difference of Doppler frequency shift is very small, so this phenomenon can be barely observed for a transmitter with larger bandwidth.

To prevent potential collisions or debris for future missions, the mission of Longjiang-2 ended with a planned lunar impact on 31 July 2019, as a result of a maneuver performed on 24 January 2019 to lower the periapsis of the satellite and the orbital perturbations over time. During the whole mission, the VHF/UHF radios have been activated for 177 times. 20945 GMSK packets and 883 JT4G packets were collected by 50 different ground stations from 17 countries, including the Netherlands, Germany, Japan, Spain, the US, Mauritius, Israel, Chile, UK, Italy, Argentina, Denmark, Brazil, Poland, Australia, Latvia, and China. In all, 763 uplink commands were sent via VHF from one ground station in Germany (the ground station of radio amateur Reinhard Kühn DK5LA) and two ground stations in China (the ground station of radio amateur Zhang Jianhua BA7KW and the ground station of Harbin Institute of Technology). In total, 135 images taken by the miniature CMOS camera were fully or partially download.

**Performance analysis of downlink signals**. A study of the performance of the communications system has been made using IQ

data gathered by four stations around the world: Dwingeloo in the Netherlands, Wakayama in Japan, and Shahe and Harbin in China. Information of the ground stations used is shown in Supplementary Table 4. The data was recorded on 5 July 2019 from 07:40 to 08:00 UTC and consists of a 500-baud GMSK transmission at 436.400 MHz sending a single image in SSDV format. An $r = 1/4$ turbo code was used as FEC. Below we show an evaluation of some key parameters of the communications system: spectrum, ASM detection and constellation diagram. These measure the performance of the whole communications chain, including the transmitter, propagation path, ground station, and software receiver.

The spectrum of the GMSK signal, as received in each of the ground stations, is shown in Fig. 4e. The signal-to-noise ratio (SNR) of the signal received in each ground station is different, owing to their different antennas.

The figures in the first column of Fig. 5 show the correlation of the signal with the ASM, in the FFT bin where the main peak is detected. As described in Section 3, the ASM detection algorithm works by multiplying the signal with the complex conjugate of the ASM and taking an FFT to search in frequency and integrate coherently for the complete duration of the ASM. In the figures the signal amplitude is normalized so that the noise power in each of the FFT bins is one when the signal is not present. Note that the correlation of the signal with the ASM is significantly larger than one even well away from the main peak. This is caused by the transmitted data having a nonzero cross-correlation with the ASM.

Additionally, the amplitude and frequency of the main peak is evaluated in the figures in the second column of Fig. 5. The "correlation (with scalloping loss)" trace shows the magnitude of the main peak in the FFT bin where the power is largest. Some of the signal power is thus lost to other FFT bins due to scalloping loss. The "correlation (no scalloping loss)" trace sums over several FFT bins instead to recover most of the signal power. Hence, it gives a good estimate of the signal $E_b/N_0$.

We make the following remarks about the ASM figures. First, we see that the frequency at which the ASM is detected keeps decreasing steadily due to changing Doppler, but there is a jump of some 20 Hz in the middle. This was caused by an occasional frequency jump in the temperature compensate crystal oscillator (TCXO) of the Longjiang-2 transmitter, as the compensation was done by digital switching. This problem was observed during ground test, but we did not have enough time to find a replacement. During spacecraft operations, these occasional jumps of the TCXO corrupted some of the received packets, since they made the receiver PLL lose lock. Second, the data recorded at Harbin shows a reliable detection of the ASM for a weak signal of ~−2 dB $E_b/N_0$. Since decoding the turbo coded data requires an $E_b/N_0$ ~0 dB, this shows that the ASM detection is robust enough.

By examining the symbols at the output of the OQPSK demodulator, we can make the constellation plots, as shown in the third column of Fig. 5. A PLL bandwidth of 8 Hz was used in this analysis. The data frames used to draw the plots have no error at turbo decoder output. As the SNR decrease, the constellation points become larger due to the additive noise. If one of the points is misidentified as another, a symbol error happens. The SNR in Harbin was low enough that the symbols were no longer recognizable in the constellation plot. However, the turbo decoder was still able to recover valid frames. This shows that the receiving system has good performance even at very low SNR.

**First lunar orbit UHF VLBI experiment**. Very-long-baseline interferometry (VLBI) is an interferometry technique used in

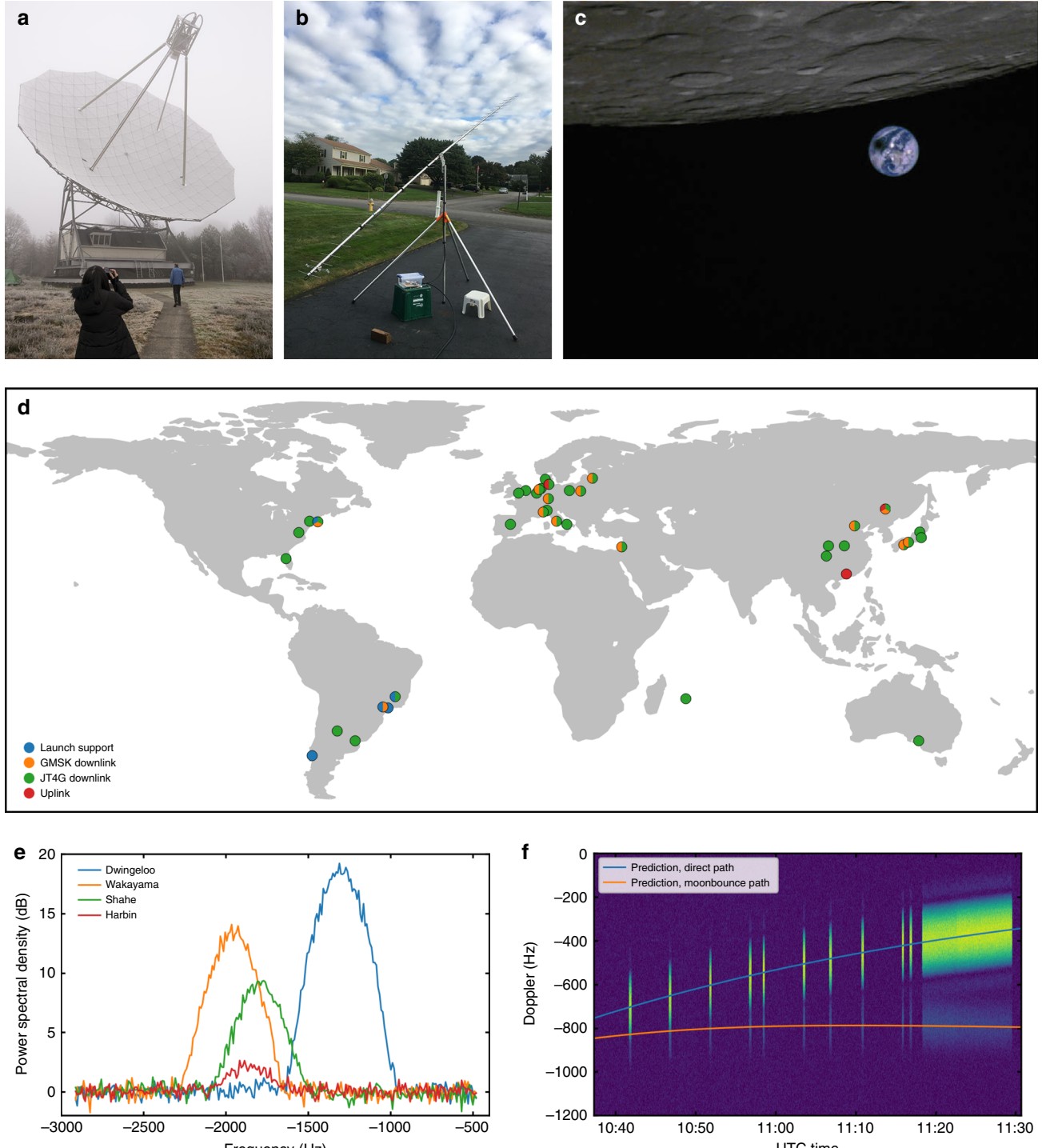

**Fig. 4 Operation of VHF/UHF radio. a** Dwingeloo 25 m radio telescope (PI9CAM) in the Netherlands, as the largest participating ground station antenna. **b** 28-element Yagi antenna on a tripod, used by radio amateur Robert Mattaliano (N6RFM) in the US, as an example of a small ground station antenna. **c** Image of the 2 July 2019 total solar eclipse (color corrected). **d** Positions of the ground stations taking part in the operation of the VHF/UHF radio. **e** Spectrum plot of GMSK downlink received by different ground stations. **f** Waterfall plot of GMSK downlink from direct path and moonbounce path comparing with prediction.

radio astronomy and spacecraft orbit determination. In VLBI a signal from a radio source is collected at multiple synchronized ground stations separated by a few hundred or thousand kilometers. The time difference between the arrivals of the radio signal at different ground stations is measured to locate the direction or position of the radio source. As the baseline is very long compared to the wavelength of the radio signal, the measurement can be quite accurate. The frequency difference can also be measured to determine the velocity of the radio source. The block diagram of signal processing used for Longjiang-2 VLBI is shown in Fig. 6a.

The synchronization between distant receivers was historically quite difficult to achieve. Nowadays it is much easier with the help of GPS. Each UHF VLBI ground station of Longjiang-2 has a

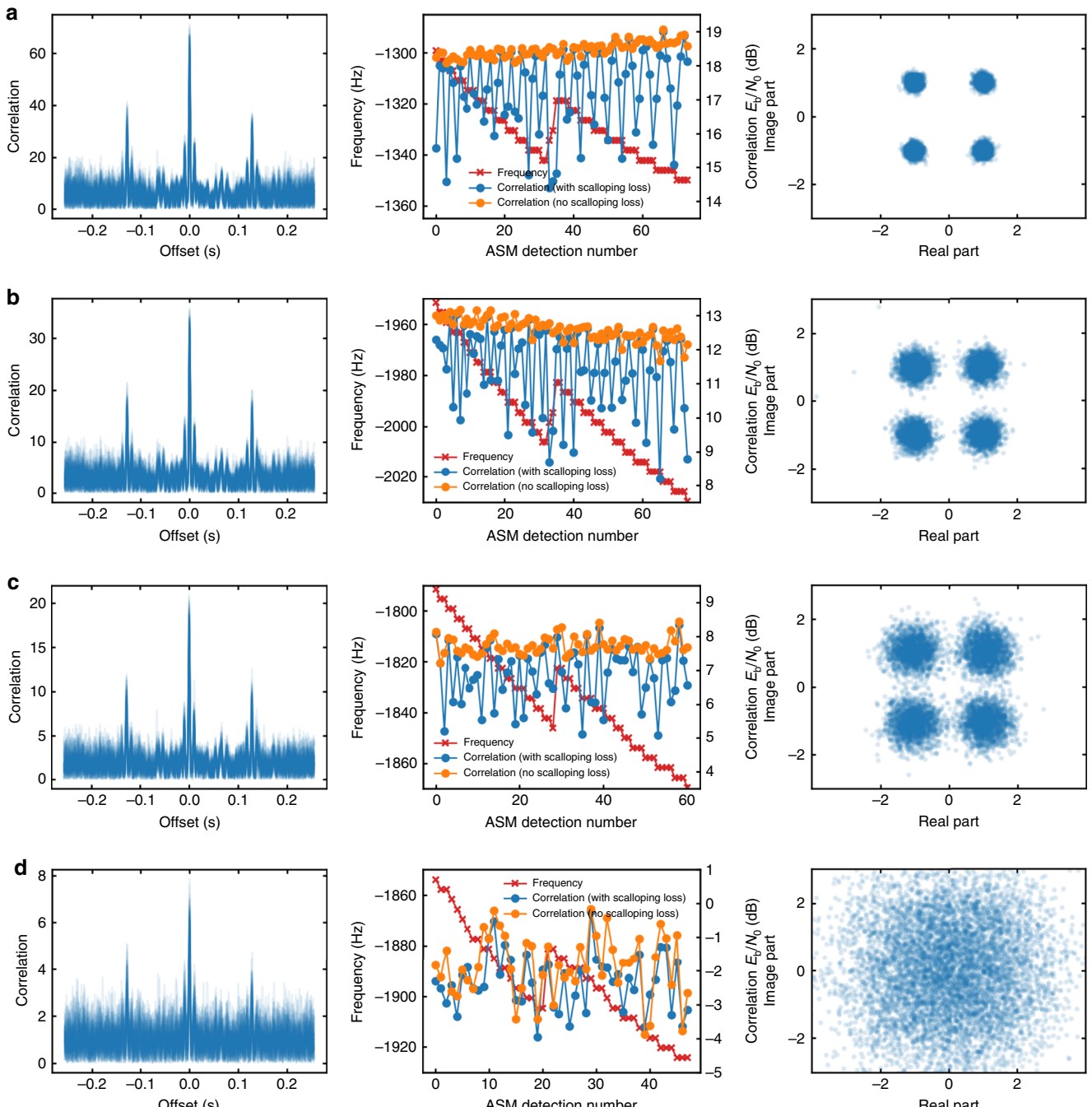

**Fig. 5 Downlink performance of Longjiang-2 received by different ground stations.** Correlation peaks, ASM detected, and constellation plots received at **a** Dwingeloo, **b** Wakayama, **c** Shahe, and **d** Harbin.

GPS disciplined oscillator (GPSDO) to provide a 10-MHz reference output with a Allan deviation of $1 \times 10^{-11}$ at 1 s, and a pulse per second (PPS) signal output with an accuracy to UTC of ±50 ns RMS. The local oscillator of the receiver is locked to the 10 MHz reference, and the sample moment of the receiver is marked referenced to the PPS signal.

The first synchronized I/Q recordings of Longjiang-2 UHF downlink were made in Dwingeloo and Shahe from UTC 04:20 to 5:40 on 10 June 2018. It was the first VLBI experiment with a lunar orbit spacecraft operating on UHF band. The distance between the ground stations is ~7250 km. The satellite was transmitting 250 bps GMSK with $r = 1/2$ turbo code in burst mode on 435.4 MHz and 436.4 MHz. The recordings are at 40 ksps sample rate centered at these two frequencies.

As Longiang-2 has both VHU/UHF and S-band radios, the orbital elements (as shown in Supplementary Table 5, measured by CDSN with S-band two-way ranging, which have a known position error of no more than 10 km) can be used to evaluate the performance of UHF VLBI. Figure 6b–e shows the delta-range and delta-velocity results of the VLBI observation. The curves fit the prediction from CDSN elements quite well, with delta-range residuals of 29.23 km RMS and 17.84 km RMS on 435.4 MHz and 436.4 MHz, and delta-velocity residuals of 0.1406 m/s RMS and 0.1437 m/s RMS on 435.4 MHz and 436.4 MHz.

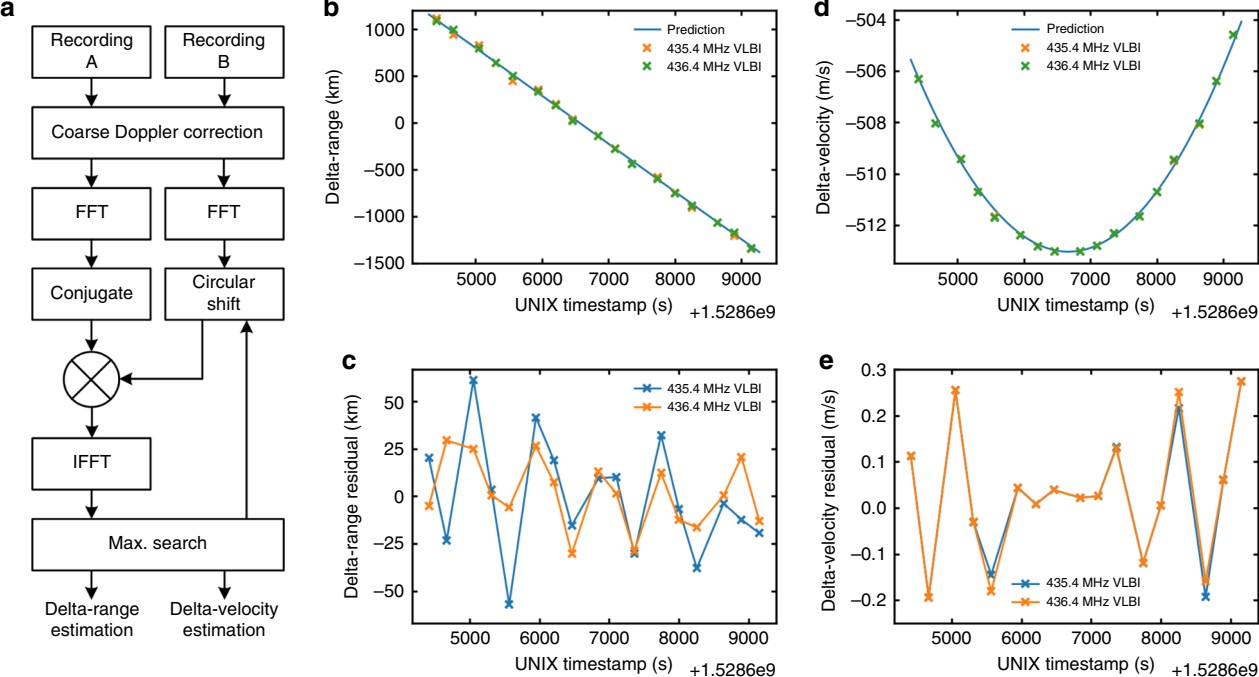

**Fig. 6 Results of the first VLBI experiment with Longjiang-2. a** Block diagram of VLBI signal processing. **b** Delta-range result from VLBI comparing with prediction from CDSN elements. **c** Delta-range residual result from VLBI. **d** Delta-velocity result from VLBI comparing with prediction from CDSN elements. **e** Delta-velocity residual result from VLBI.

## Discussion

Despite the regretful loss of Longjiang-1, the mission of the VHF/UHF radio of Longjiang-2 has been a great success. It was the first amateur radio communication system operating in lunar orbit, and provided a lot of data return during its 14 months lifetime. With the help of new hardware and waveform design, the VHF/UHF radio provides excellent performance at the cost of limited available weight, power, and envelope resources, enabling the use with small ground stations and simple hardware, making it the lunar mission with most ground stations involved. The first lunar orbit VLBI experiment on UHF band was also carried out based on the VHF/UHF radio. The concepts and techniques developed can be used for the communication system design of future miniature or low-cost deep space missions.

## Data availability

Datasets for telemetries from Longjiang-1/2 are available in the DSLWP public data release v1.0 repository, https://doi.org/10.5281/zenodo.3571330. Datasets for the raw signal recordings from Longjiang-2 are available in the CAMRAS DSLWP data repository, https://charon.camras.nl/public/dslwp-b. Other data that support the paper and other findings of this study are available from the corresponding authors upon reasonable request.

## Code availability

The code that supports this paper (including modulator, demodulator, correlator, plot drawing, etc.) is available in the gr-dslwp repository, https://github.com/bg2bhc/gr-dslwp.

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

## Acknowledgements

This research was a part of the Chang'e-4 mission of Chinese Lunar Exploration Program, funded by the government of Heilongjiang Provence and Harbin Institute of Technology. Beijing Aerospace Control Center, Tianlai team of National Astronomical Observatories, CAS, State Radio Monitoring Center and radio amateur community all over the world provided unselfish support during the operation of the VHF/UHF radio. We thank all the people involved for their excellent work and efficient cooperation.

## Author contributions

M.W., X.C., and F.W. generated the design concept. M.W. designed the overall system architecture and transceiver hardware. M.W. and D.E. designed and analyzed the performance of the waveforms. C.H. designed and tested onboard antennas and ground station antennas in Harbin and Shahe. M.T. designed and tested the CMOS camera. Y.Z. designed the ground station software. C.B. and T.J.D. led the observations in Dwingeloo and spotted the moonbounce propagation. D.E., M.W., and J.H. designed the VLBI observations and processing. M.W., C.H., and D.E. wrote the manuscript. X.C. and F.W. supervised the project. All authors reviewed and commented on the manuscript.

## Competing interests

The authors declare no competing interests.
