## [Peer Review File · Nature Communications]

Reviewer #2 (Remarks to the Author):

Paper Summary

The author of this paper describes and elaborates about the design and implementation of the VHF/UHF radio and the waveforms used by China's Chang'e-4 lunar far side mission. Moreover, the paper reports about the mission which was composed of two lunar microsatellites for low frequency radio astronomy, amateur radio and education, together with the Queqiao L2 relay satellite. The communication system operated in lunar orbit by means of a VHF/UHF SDR for small ground stations. The authors also provide downlink performance analysis and interferometry experiment.

Comments

The paper is very interesting, solid and sound and provides exhaustive results.

Nonetheless, the reviewer deems that the SotA can be more effectively referred to by adding more references to Small sats. To this aim, the reviewer suggests to cite references such [1-2] in the introduction part.

Paper Recommendation

On the reviewer's opinion, the paper can be accepted even if an answer to the minor point that is defined in the review would be beneficial.

References

[1] Burleigh S. C. et alii: "From Connectivity to Advanced Internet Services: A Comprehensive Review of Small Satellites Communications and Networks", *Wireless Communications & Mobile Computing (WCMC) Journal*, Special Issue on Emerging Multiple Access Technologies, Vol 2019.

[2] F. Davoli et alii: "Small satellites and CubeSats: Survey of structures, architectures, and protocols," *International Journal of Satellite Communications and Networking*, 2018.

Reviewer #3 (Remarks to the Author):

Well crafted paper that described in detail the design and results of the communication system on the Longjiang-1 and 2 spacecraft. Some comments:

- Line 5, item b) on page 3: Did you exercise this option to validate it could work in the near Earth orbital region?
- Line 41 on page 3: Did you identify the IF in the paper? I thought I read something about 70kHz, but I couldn't find it again.
- Line 10, right hand paragraph, believe the word "satellite" should be capitalized.
- Line 15 on page 4, right hand paragraph: It was not apparent why the FER results are different for Longjiang-2 vs 1. What was the physical difference between the two satellites? Or is this referring to Longjiang-2 achieving lunar orbit thus higher path loss?
- Line 42 on page 6, the reference to anti-jamming improvement did not identify a reference point, from where it was improved. The assumption here is that its better, but not quantified, and the value of improvement supports hacking to the signals while in Lunar orbit.
- Line 5 page 6, did you validate that the status data of the radio that was sent down made sense? That is, if it was temperature or power usage, you could validate the numbers were reasonable, and thus all working correctly? A typical use of a beacon is also to get some health and status of the satellite itself in the case of the larger downlink not operating. Was this not included?
- Line 22, page 6 right hand paragraph: You reference overheating of batteries, can you provide any other context? Was this due to the UHF system operating or to multiple other systems on board operating? Just curious if this points to higher than average power draw from the RF system.
- Figure 4c is a nice output from the camera onboard looking back. Given the very low data rate of the UHF, can you identify how long it took to send all of this images data, or how many passes it took to take it out of memory and then send along?
- Line 22 on page 8, right hand paragraph: Can you explain why the 20Hz jump occurred to the TCXO? Do you think this was a component issue onboard?

- Line 39 on page 8, right hand paragraph; It is uncertain if you can claim that the receiving system had good performance at very low SNR; given that the signal was very low at Harbin you could not pull out distinguishable I/Q points. Could you quantify "very low" referring to a PSD that conforms to what was received at Shahe? That was coherent and recognizable, albeit a bit noisy. How far down in PSD is that from Dwingeloo that had the best? No argument the system has good performance, just what is meant by "good" is relative to what is received from the best performing ground antenna.

- Line 8 on page 10: Can you provide some quantitative output of the VLBI results? The oscillators has good signal output accuracy, thus were you able to correlate the resultant signals to a data parameter sent from the Moon? For future study this would be interesting to understand what is possible on the surface for science sensing.

16th Apr 2020

Response to Referees

Dear Referees,

Thank you very much for your positive reports for our manuscript "Design and Flight Results of the VHF/UHF Communication System of Longjiang-1/2 Lunar Microsatellites". In this letter, we provide a point-by-point response to your comments. All changes in the manuscript are marked with yellow highlighting.

We deeply appreciate your consideration of our manuscript, and we look forward to receiving your further comments.

Reviewer #2 (Remarks to the Author):

Paper Summary

The author of this paper describes and elaborate about the design and implementation of the VHF/UHF radio and the waveforms used by China's Chang'e-4 lunar far side mission. Moreover the paper reports about the mission which was composed by two lunar microsatellites for low frequency radio astronomy, amateur radio and education, together with the Queqiao L2 relay satellite. The communication system operated in lunar orbit by means of a VHF/UHF SDR for small ground stations. The authors also provide downlink performance analysis and interferometry experiment.

Comments

The paper is very interesting, solid and sound and provides exhaustive results. Nonetheless, the reviewer deems that the SotA can be more effectively referred by adding more references to Small sats. To this aim the reviewer suggest to cite references such [1-2] in the introduction part.

Paper Recommendation

On the reviewer's opinion, the paper can be accepted even if an answer to the minor point that is defined in the review would be beneficial.

References

- [1] Burleigh S. C. et alii: "From Connectivity to Advanced Internet Services: A Comprehensive Review of Small Satellites Communications and Networks", Wireless Communications & Mobile Computing (WCMC) Journal, Special Issue on Emerging Multiple Access Technologies, Vol 2019.
- [2] F. Davoli et alii: "Small satellites and CubeSats: Survey of structures, architectures, and protocols," International Journal of Satellite Communications and Networking, 2018.

We cited the references in line 4 on page 1, right column of the updated version.

Reviewer #3 (Remarks to the Author):

Well crafted paper that described in detail the design and results of the communication system on the Longjiang-1 and 2 spacecraft. Some comments:

- Line 5, item b) on page 3: Did you exercise this option to validate it could work in the near Earth orbital region?

Before Longjiang-1/2, we have developed several LEO satellites with VHF/UHF radios, mentioned a few paragraphs later (line 17 on page 3, right column of the updated version). For Longjiang-1/2, they did not have an earth orbiting phase as the launch vehicle directly sent them to a lunar transfer orbit. But the radios were powered on at separation, so we were able to test the radios during the first a few hours of LTO, and it was successful, as mentioned in line 21 on page 6, right column of the updated version.

We made no modification to the paper for this comment.

- Line 41 on page 3: Did you identify the IF in the paper? I thought I read something about 70khz, but I couldn't find it again.

The IF is 98kHz, mentioned a few paragraphs later (line 23 on page 3, right column of the updated version): "then down-converted to 98 kHz intermediate frequency (IF) by an image rejecting I/Q demodulator".

We made no modification to the paper for this comment.

- Line 10, right hand paragraph, believe the word "satellite" should be capitalized.

Yes, this was a typo. We replace "satellite ARISSat-1" with "The satellite ARISSat-1".

- Line 15 on page 4, right hand paragraph: It was not apparent why the FER results are different for Longjiang-2 vs 1. What was the physical difference between the two satellites? Or is this referring to Longjiang-2 achieving lunar orbit thus higher path loss?

The design and consideration for Longjiang-1 and Longjiang-2 is the same, and more difficult than a standard DSN OQPSK receiver, as the signal is weaker and the

bandwidth is narrower, an ASM detector was designed. The BER plot shows difference between the LRTC demodulator and some other typical demodulators, not the difference for Longjiang-2 vs 1.

We changed the expression "Longjiang-2" by "Longjiang-1/2" in line 5 and line 6 on page 5, right column of the updated version.

- Line 42 on page 6, the reference to anti-jamming improvement did not identify a reference point, from where it was improved. The assumption here is that its better, but not quantified, and the value of improvement supports hacking to the signals while in Lunar orbit.

The ASM detector is a correlator for a pseudo-random sequence. It provides processing gain, typically 15 dB for a 32-bit pseudo-random sequence, that attenuates several kinds of interfering signals, comparing to systems that do not use DSSS. But the actually performance depends on the type of interfering signal.

We changed the expression "As the ASM detector is a correlator for a pseudo-random sequence, the anti-jamming performance is improved." by "The ASM detector is a correlator for a pseudo-random sequence. It is known that this introduces processing gain that attenuates several kinds of interfering signals."

- Line 5 page 6, did you validate that the status data of the radio that was sent down made sense? That is, if it was temperature or power usage, you could validate the numbers were reasonable, and thus all working correctly? A typical use of a beacon is also to get some health and status of the satellite itself in the case of the larger downlink not operating. Was this not included?

Yes, the data sent down were used to monitor the spacecraft's health and performance, for example the temperature of onboard batteries (mentioned in the next comment). Here is a link to the telemetry parser page of Longjiang-2, showing the temperature curve during the last activations:

http://lilacsat.hit.edu.cn/dashboard/pages_en/lines.html?sat=DSLWP-B&data=WOD-C00074.7.16

The datasets for telemetries (beacon included) from Longjiang-1/2 are available in the DSLWP public data release v1.0 repository, <https://doi.org/10.5281/zenodo.3571330>, we had included this information in "Data availability" section. We also added some

description for the modes of GMSK telemetry (including a burst mode as a beacon, and a stream mode) to line 8 on page 4, right column of the updated version.

- Line 22, page 6 right hand paragraph: You reference overheating of batteries, can you provide any other context? Was this due to the UHF system operating or to multiple other systems on board operating? Just curious if this points to higher than average power draw from the RF system.

The power drawn from the RF system was as expected. The overheating was the result of the unexpected heat dissipation of a diode in series with the solar panel. During thermal vacuum test, we powered the satellite with ground power supply, not the solar panel, so the power dissipation was not presented, and the thermal control system was designed based on this result. After launch, when the solar panel was illuminated, the temperature of onboard Li-ion batteries was higher than expected by about 8°C, as the diode was just mounted next to the batteries. As the position of the VHF/UHF radio was also quite near to the batteries, when the radio was on, the temperature conditions of the batteries got worse. As a result, we limited the time slot to activate the VHF/UHF radio.

We did not add this information to the updated version of the paper to avoid increasing its length a lot.

- Figure 4c is a nice output from the camera onboard looking back. Given the very low data rate of the UHF, can you identify how long it took to send all of this images data, or how many passes it took to take it out of memory and then send along?

An image took typically 10 to 30 minutes to transmit. The spacecraft was visible almost anytime that the Moon was visible. However, the UHF transmitter was typically operated in 2-hour slots (at most one or two per day) to prevent overheating. In each of these slots, it was possible to download several images.

We added this information to the updated version of the paper by replacing "and downloaded via the VHF/UHF radio on 3 July 2019" with "and transmitted via the VHF/UHF radio using slow scan digital video (SSDV) format on 3 July 2019. The file size of the image is 19.1 kbytes and took about 22 minutes to download at 500 baud with $r=1/4$ Turbo code". We also added reference 19 for the format of image transmission.

- Line 22 on page 8, right hand paragraph: Can you explain why the 20Hz jump occurred to the TCXO? Do you think this was a component issue onboard?

It is caused by the onboard TCXO, but it is normal, it's not a faulty component. Some TCXOs do digital switching for temperature compensation (like switching in capacitors of different values), this makes the output frequency jump a little. Some TCXOs do this, others don't. This problem was observed during ground test, but we did not have enough time to find a replacement.

We added this information to the updated version of the paper by replacing "This was caused by a frequency jump in the TCXO of the Longjiang-2 transmitter." with "This was caused by an occasional frequency jump in the temperature compensate crystal oscillator (TCXO) of the Longjiang-2 transmitter, as the compensation was done by digital switching. This problem was observed during ground test, but we did not have enough time to find a replacement."

- Line 39 on page 8, right hand paragraph; It is uncertain if you can claim that the receiving system had good performance at very low SNR; given that the signal was very low at Harbin you could not pull out distinguishable I/Q points. Could you quantify "very low" referring to a PSD that conforms to what was received at Shahe? That was coherent and recognizable, albeit a bit noisy. How far down in PSD is that from Dwingeloo that had the best? No argument the system has good performance, just what is meant by "good" is relative to what is received from the best performing ground antenna.

We did claim that the receiving system had good performance at very low SNR because of the results at Harbin. During the pass on 5 July 2019, the SNR at Harbin is very critical (around the threshold), we could not pull out distinguishable I/Q points, but with the help of the ASM detector and turbo decoder, we can still detect valid frames with no bit errors (confirmed by checking the CRCs).

To state it more clearly, we modified "The constellation points from Harbin cannot even be distinguished from each other, so many symbol errors are produced at the demodulator output, but are all corrected by the Turbo decoder." by "The SNR in Harbin was low enough that the symbols were no longer recognizable in the constellation plot. However, the Turbo decoder was still able to recover valid frames."

- Line 8 on page 10: Can you provide some quantitative output of the VLBI results? The oscillators has good signal output accuracy, thus were you able to correlate the resultant signals to a data parameter sent from the Moon? For future study this would be interesting to understand what is possible on the surface for science sensing.

The main purpose of the VLBI experiment was to determine the orbit of the satellite, so we showed some results of delta-range and delta-velocity measurements in Fig 6. The quantitative analysis of the results was shown in the last paragraph of this section, "with delta-range residuals of 29.23 km RMS and 17.84 km RMS on 435.4 MHz and 436.4 MHz, and delta-velocity residuals of 0.1406 m/s RMS and 0.1437 m/s RMS on 435.4 MHz and 436.4 MHz".

Thank you very much for your kind reminder for future study. In Fig 4f we mentioned that the UHF downlink signals bounced off the Moon was observed during some of the activations. We are trying to correlate the bounced signal with the directly propagated signal to see if we can get some geographic information of lunar surface. As the reflection was very weak, we have to process more than 200 GB of data to collect useful portions. This is out of the scope of this paper, so we are planning to cover this in our next one. At this moment, we made no modification to the manuscript.

If you have any queries, please don't hesitate to contact me at the address below.

Hope you and your family stay safe and healthy!

Yours sincerely,

Prof. WANG Feng (corresponding author)

School of Astronautics

Harbin Institute of Technology

E-mail: wfhitsat@hit.edu.cn

Tel: 0086-86416447

Fax: 0086-86416457

REVIEWERS' COMMENTS:

Reviewer #2 (Remarks to the Author):

On the reviewer's opinion, the paper can now be accepted in its present form.

Reviewer #3 (Remarks to the Author):

The updates and answers provided were concise and all made sense.

31st May 2020

Response to Referees

Dear Referees,

Thank you again for your review for our manuscript "Design and Flight Results of the VHF/UHF Communication System of Longjiang Lunar Microsatellites".

In this letter, we provide a point-by-point response to your comments. All the changes in the main text were made using the "track changes" feature. We deeply appreciate your consideration of our manuscript.

Reviewer #2 (Remarks to the Author):

On the reviewer's opinion, the paper can now be accepted in its present form.

Reviewer #3 (Remarks to the Author):

The updates and answers provided were concise and all made sense.

Thank you very much for your positive feedback. We made no modification to the paper for these comments.